# A Palladium-Catalyzed 4CzIPN-Mediated Decarboxylative Acylation Reaction of O-Methyl Ketoximes with α-Keto Acids under Visible Light

**DOI:** 10.3390/molecules27061980

**Published:** 2022-03-18

**Authors:** Cheng Wang, Shan Yang, Zhibin Huang, Yingsheng Zhao

**Affiliations:** 1Key Laboratory of Organic Synthesis of Jiangsu Province, College of Chemistry, Chemical Engineering and Materials Science Soochow University, Suzhou 215123, China; 20204209201@stu.suda.edu.cn (C.W.); 20184209164@stu.suda.edu.cn (S.Y.); 2School of Chemistry and Chemical Engineering, Henan Normal University, Xinxiang 453000, China

**Keywords:** C–H activation, decarboxylation, palladium catalysis, visible light

## Abstract

This work presents a palladium-catalyzed oxime ether-directed ortho-selective benzoylation using benzoylformic acid as the acyl source with a palladium catalyst and 4CzIPN as the co-catalyst under light. Various non-symmetric benzophenone derivatives were obtained in moderate to good yields. A preliminary mechanism study revealed that the reaction proceeds through a free radical pathway.

## 1. Introduction

Diaryl ketones are important scaffolds that exist in a wide range of biologically active molecules and drugs, such as ketoprofen, fenofibrate, etc. (Figure 1) [1].

Transition-metal-catalyzed decarboxylative coupling is an attractive synthetic method for C–C bond formation since it does not require the use of stoichiometric organometallic coupling reagents and forms CO_2_ as a byproduct instead of toxic metal compounds [2]. In 2008, a general study was first reported on the synthesis of non-symmetrical ketones from α-oxocarboxylates and aryl bromides (Figure 1a) [3]. Since then, various scientists have conducted extensive investigations in this area [4,5,6]. In 2010, Ge first reported a room-temperature, decarboxylative ortho-acylation of acetanilides with α-keto acids via palladium-catalyzed C–H activation [7]. Subsequently, the ortho-acylation of various substrates with α-keto acids via metal-catalyzed C–H activation was developed [8,9,10,11]. However, it has been found that the above methods often require the use of excessive persulfate oxidizer or silver salt to promote the decarboxylation of α-keto acids, which limits their further synthetic application. Herein, a palladium-catalyzed oxime ether-directed ortho-benzoylation reaction under light conditions with 4CzIPN as the photocatalyst and benzoylformic acid as the acyl source is reported, with O_2_ as the only oxidant. A wide variety of non-symmetric benzophenone compounds can be easily obtained with moderate to good yields by this route.

In 2014, Lei and co-workers developed a cross-coupling reaction of benzoylformic acid and aniline in the presence of a photocatalyst, oxygen atmosphere and light (Figure 1b) [12]. The mechanism studies revealed that the reaction proceeded through a benzoyl radical intermediate. Inspired by these results, it was speculated that the indispensable oxidants such as excessive persulfate or silver salts may be replaced by using a photocatalyst with light, which would open a new, highly efficient pathway for decarboxylative reactions. Recently, our group presented several examples of oxime ether-directed selective C–H functionalization [13,14,15]. It is envisioned that when acetophenone oxime ether is employed as the substrate and benzoylformic acid as the source of benzoyl, the use of a photocatalyst under light may achieve direct ortho-selective benzoylation with a palladium catalyst.

## 2. Results and Discussion

### 2.1. Optimization of the Reaction Conditions

With these conditions in mind, acetophenone oxime ether (**1a**) was first treated with benzoylformic acid (**2a**) using Pd(OAc)_2_ (10 mol%) and Eosin Y (2 mol%) as the catalysts and PhCl (0.2 M) as the solvent at r.t. under the irradiation of 40-watt blue LEDs for 18 h (Table 1, entry 1). Fortunately, the ortho-benzoylated acetophenone oxime ether **3a** was obtained in 55% yield. Then, other palladium catalysts were tested, such as Pd(TFA)_2_ and PdCl_2_ (Table 1, entries 2–3), but failed to achieve better results. Next, photocatalysts such as fac-Ir(ppy)_3_, Rose Bengal and 4CzIPN (2,4,5,6-tetra(9H-carbazol-9-yl)isophthalonitrile) were screened (Table 1, entries 4–6). The photocatalyst 4CzIPN greatly improved the transformation, leading to the desired product **3a** in 70% yield. To further improve the efficiency of this reaction, several solvents, including CH_3_CN, DCE and acetic acid, were tested (Table 1, entries 7–9). An isolated yield of 77% was achieved with acetic acid as the solvent. When 3 equiv. of benzoylformic acid was used, a satisfactory yield of **3a** was observed (Table 1, entry 10). Several control experiments were carried out (Table 1, entries 11–14), and the results clearly showed that Pd(OAc)_2_ and light were both indispensable for this reaction. Interestingly, when the photocatalyst was removed from the reaction, **3a** was also obtained with a yield of 30%, which suggested that the photocatalyst only promoted the generation of benzoyl radicals.

### 2.2. Scope of Benzoylformic Acid

With the optimal conditions in hand, the applicability of benzoylformic acid substrates was investigated next (Figure 2). Benzoylformic acids substituted at the para-position, such as fluorine, chlorine, bromine and methoxy, were well tolerated, yielding the corresponding products in good to excellent yields (**3a**–**3e**). The meta-methoxy, fluoro and chloro benzoylformic acids also provided the target products (**3f**–**3h**) in moderate yields. Ortho-methyl and bromo-substituted benzoylformic acids were both well compatible in this reaction, leading to the corresponding products **3i** (60%) and **3j** (21%), respectively. It was found that 3,4-(methylenedioxy)benzoyl formic acid was also suitable for this system, generating the corresponding product **3k** with a yield of 47%. When the reaction was carried out with p-hydroxybenzoyl formic acid, the desired target product was not obtained.

Reaction conditions: **1a** (0.1 mmol), **2** (3 equiv.), Pd(OAc)_2_ (10 mol%), 4CzIPN (2 mol%), AcOH 0.2 M, air, 18 h. Isolated yields are given.

### 2.3. Scope of Acetophenone Oxime Ether

Next, the applicability of acetophenone oxime ether substrates was investigated under the optimal conditions (Figure 3). For alkyl-substituted acetophenone oxime ethers such as methyl, ethyl, isopropyl, tert-butyl or methoxy derivatives, both para- and meta-substituents yielded the corresponding products in moderate to good yields (**4a**–**4h**). Halogen-substituted acetophenone oxime ethers such as fluorine, chlorine and bromine derivatives also provided the target products (**4i**–**4p**) in moderate yields. For the meta-fluorinated acetophenone oxime ether, a mixture of the C-2 benzoylated product **4o′** and the C-6 benzoylated product **4o** was obtained. However, the ortho-fluorinated acetophenone oxime ether did not yield the target product under the standard conditions. When the reaction temperature was increased to 60 °C, the target product **4m** was obtained with an isolated yield of 23%. For the para-trifluoromethyl-substituted acetophenone oxime ether, the target product **4p** was obtained in 25% yield when the reaction temperature of the system was r.t. and the reaction time was extended to 24 h. When the temperature was increased to 60 °C, the product **4p** was obtained with a yield of 40%. In addition, 1-tetralone oxime ether was also found to be suitable for this reaction, and the corresponding product **4q** was obtained with a yield of 59%. The chroman-4-one oxime ether also delivered the target product **4r** with an isolated yield of 71%. Benzophenone oxime ether was also compatible with this reaction, and the target product **4s** was obtained with an isolated yield of 70%, which undoubtedly expanded the substrate scope. For 4,4′-disubstituted benzophenone oxime ethers with methyl, methoxy and fluorine substituents, the corresponding products (**4t**–**4v**) were obtained in moderate isolated yields. Derivatives of ketoprofen, a drug with anti-inflammatory and analgesic effects, were also successfully applied to this reaction, and a mixture of **4w** and **4w′** was obtained with a combined yield of 58% (**4w**:**4w′** = 1.075:1).

Reaction conditions: **1** (0.1 mmol), **2a** (3 equiv.), Pd(OAc)_2_ (10 mol%), 4CzIPN (2 mol%), AcOH 0.2M, air, 18 h. Isolated yields are given.

### 2.4. Gram-Scale Reaction and Further Transformation

To further test the application value of this route, a gram-scale reaction was carried out for 48 h, and the product **3a** was isolated in 72% yield (Figure 4). Then, **3a** and **4s** were chosen as representative examples to remove the directing group, and the corresponding derivatives **5** and **6** were isolated with 69% and 89% yield, respectively. Notably, the products **5** and **6** were easily transformed into phthalazine derivatives **7** and **8** in 91% and 96% yields by reaction with hydrazine hydrate. Then, **5** was treated with sodium hydroxide in water and the product **9** was obtained with an isolated yield of 71%.

### 2.5. Preliminary Mechanistic Study

Next, several control experiments were carried out to further understand the reaction pathway of this transformation (Figure 5). First, when the substrate **1a** was treated under standard conditions with 3 equiv. of 2,2,6,6-tetramethylpiperidine N-oxide (TEMPO), almost no **3a** was detected, but product **10** was obtained with a yield of 19% (Figure 5a). When **2a** was directly treated with 4CzIPN in the presence of 2 equiv. of TEMPO in AcOH under the irradiation of 40-watt blue LEDs, the final isolated yield of **10** was 58% (Figure 5b). Then, a kinetic isotope effect (KIE) experiment was conducted using p-methylacetophenone oxime ether **1b** and 2,3,5,6-tetradeutero-p-trideuterium methyl acetophenone oxime ether **1b′** as the substrates under optimal conditions. After treatment, a mixture of the products **4a** and **4a′** was obtained (Figure 5c). The K_H_/K_D_ = 2.26, which indicated that C–H bond cleavage was the rate-determining step of the reaction.

Thus, a possible reaction route was proposed based on the existing literature [16,17,18] and the aforementioned experimental results (Figure 6). At first, 4CzIPN was promoted to its excited state (4CzIPN)* under visible light irradiation. The (4CzIPN)* oxidized benzoylformic acid into free radical cations and was simultaneously reduced to (4CzIPN)^•−^. Subsequently, electron transfer from (4CzIPN)^•−^ by molecular oxygen regenerated 4CzIPN for the next run and produced a superoxide radical anion (O_2_^•−^), which was confirmed by electron spin resonance (ESR; see Figure 2). At the same time, palladium acetate activated the ortho C–H bond of the acetophenone oxime ether to obtain the five-membered ring palladium complex **I**, followed by the addition of **I** and benzoyl radicals to obtain the intermediate **II**. The superoxide radical anion oxidized **Ⅱ** to Pd (Ⅳ) intermediate **Ⅲ**; then, **Ⅲ** was eliminated by reduction to obtain the target product ortho-benzoylated acetophenone oxime ether and palladium acetate to complete the catalytic cycle.

## 3. Materials and Methods 

### 3.1. General Information

Unless otherwise noted, all reagents were purchased from Acros, Alfa and Adamas and used without further purification. Column chromatography purifications were performed using 200–300 mesh silica gel. NMR spectra were recorded on Varian Inova-400 MHz, Inova-300 MHz, Bruker DRX-400 or Bruker DRX-500 instruments and calibrated using residual solvent peaks as internal references. All heating reactions were conducted in an oil bath. Multiplicities are recorded as follows: s = singlet, d = doublet, t = triplet, dd = doublet of doublets, m = multiplet. HRMS analyses were carried out using a TOF-MS instrument with an EI source. ESR: JES-X320 electron spin resonance spectrometer.

### 3.2. Procedure for Preparation of ***2b***–***2l***

A flame-dried pressure tube was charged with the specified acetophenone (4.0 mmol, 1.0 equiv.) and selenium dioxide (0.89 g, 8.0 mmol, 2.0 equiv.). Dry pyridine (5 mL) was added, the tube was sealed, and the reaction mixture was heated at 110 °C for 18 h. Afterwards, the reaction was cooled to room temperature and filtered. The filtrate was acidified with 1 M HCl solution and extracted three times with EtOAc. Subsequently, the combined organic layers were extracted three times with 1 M NaOH solution. The combined aqueous layers were acidified with conc. HCl and extracted three times with EtOAc. The combined organic layers were dried over Na_2_SO_4_ and concentrated in vacuo. Purification by column chromatography yielded the products **2b**–**2l**. The characterization results are consistent with the references [19] (Please see Appendix A).

### 3.3. Preparation of O-Methyl Ketoximes ***1a***–***1w***

To a solution of ketones (22.0 mmol) and pyridine (5.0 mL, 61.8 mmol) in EtOH (10 mL), NH_2_OMe•HCl (2.29 g, 33.0 mmol) was added in one portion, and the reaction mixture was stirred at 60 °C for 6 h. The reaction was quenched by adding water and extracted twice with ethyl acetate. The combined extracts were washed with aqueous HCl and brine and dried over MgSO_4_. The solvents were removed under reduced pressure. Further recrystallization was conducted from ethyl acetate-hexane to provide O-methyl ketoximes [14].

### 3.4. General Procedures for Ortho-Benzoylation of Acetophenone Oxime Ether

Ketoxime ether (**1**, 0.10 mmol), aroylformic acid (**2**, 0.30 mmol), Pd(OAc)_2_ (10 mol%), 4CzIPN (2 mol%) and AcOH (0.2 M) were added to an oven-dried reaction vessel equipped with a magnetic stirring bar, and the reaction vessel was irradiated using 40 W blue LEDs at r.t. for 18 h. After the reaction was completed, the reaction solution was concentrated under reduced pressure to yield crude product **3**, which was purified by flash chromatography (silica gel, petroleum ether/ethyl acetate).

### 3.5. Gram-Scale Reaction

Acetophenone oxime ether (**1a**, 7 mmol), benzoylformic acid (**2a**, 21 mmol), Pd(OAc)_2_ (10 mol%), 4CzIPN (2 mol%) and AcOH (0.2 M) were added to an oven-dried reaction vessel equipped with a magnetic stirring bar, and the reaction vessel was irradiated using 40 W blue LEDs at r.t. for 48 h. After the reaction was completed, the reaction solution was concentrated under reduced pressure to yield crude product **3a**, which was purified by flash chromatography (silica gel, petroleum ether/ethyl acetate = 50:1).

### 3.6. Further Transformation

A 15-milliliter vial equipped with a magnetic stirrer was charged with **3a** (or **4s**) (0.2 mmol) and EtOH (1 mL). Then, 37 wt.% formaldehyde solution (1 mL) and concentrated hydrochloric acid (160 µL) were added, and the mixture was stirred and heated at 40 °C for 24 h. The reaction was diluted with ethyl acetate. The mixture was concentrated under reduced pressure and then purified by flash column chromatography to give product **5** (or **6**) [20].

To the solution of the 1,2-diacylbenzenes (**5** or **6**) (0.2 mmol) in ethanol (2 mL) was added hydrazine hydrate (1 mmol), and the resulting mixture was heated at reflux for 6 h under nitrogen. After cooling to room temperature, ethanol was removed under vacuum to give the crude product, which was purified by silica gel column chromatography (ethyl acetate, hexane and methanol) to give the phthalazines (**7** or **8**) [21].

To the solution of the 1-(2-benzoylphenyl)ethan-1-one **5** (0.2 mmol) in H_2_O (1 mL) was added sodium hydroxide (1 mmol), and the resulting mixture was heated at 20 °C for 6 h. The reaction was diluted with ethyl acetate. The mixture was concentrated under reduced pressure and then purified by flash column chromatography to give product **9** [22].

### 3.7. Radical Capture Experiment

Acetophenone oxime ether (**1a**, 0.1 mmol), benzoylformic acid (**2a**, 0.3 mmol), Pd(OAc)_2_ (10 mol%), 4CzIPN (2 mol%), 2,2,6,6-tetramethyl-1-piperinedinyloxy (0.3 mmol) and AcOH (0.2 M) were added to an oven-dried reaction vessel equipped with a magnetic stirring bar, and the reaction vessel was irradiated using 40W blue LEDs at r.t. for 18 h. After the reaction was completed, the reaction solution was concentrated under reduced pressure to yield crude product **10**, which was purified by flash chromatography (silica gel, petroleum ether/ethyl acetate = 30:1).

### 3.8. Experimental Procedure of Kinetic Isotope Effect

Pd(OAc)_2_ (0.01 mmol, 2.3 mg), 4CzIPN (0.002 mmol, 1.6 mg), **1b** (0.1 mmol,) **1b′** (0.1 mmol), benzoylformic acid (0.3 mmol) and acetic acid (0.5 mL) were added in sequence to a 5-milliliter glass reaction flask with a magnetic stir bar, and the reaction flask was irradiated under 40 W blue LEDs and stirred at r.t. for 6 h. After the reaction, the reaction solution was diluted with ethyl acetate and filtered through Celite. The filtrate was concentrated in vacuo and purified by silica gel column chromatography to obtain the target products **4a** and **4a′**. According to the 1H NMR results, we calculated K_H_:K_D_ = 2.26 (2.08 ÷ (3 − 2.08) = 2.26).

### 3.9. Determination of Superoxide Radical Anion

A superoxide radical anion (O_2_^•−^) was generated from molecular oxygen by single electron transfer (SET) [23]. We used 5,5-dimethyl-1-pyrroline-N-oxide (DMPO) as a probe to capture the active species O_2_^•−^ [24]. As shown in Figure 2, when the solution of DMPO, **2a** and 4CzIPN was in chlorobenzene solution without irradiation, no signal was detected. In contrast, when the same solution was irradiated with blue LEDs, the signal showing that O_2_^•−^ was trapped by DMPO was observed.

## 4. Conclusions

In summary, a dual catalytic system for the ortho-acylation of acetophenone oxime ether was developed herein, and various non-symmetric benzophenone derivatives were obtained in moderate to good yields. The use of a catalytic amount of the photoredox catalyst under the irradiation of blue LEDs avoided the typical high loading of external oxidants. The preliminary mechanistic analysis revealed that the ortho-acylation reaction proceeded via a free radical pathway, and the photoredox catalyst played an important role in the decarboxylation reaction. Further investigations on the combination of photoredox catalysts with transition metal catalysts for other organic transformations are currently underway in our lab.

## Data Availability

The data presented in this study are available on request from the corresponding author.

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
