# Peer review of "A Palladium-Catalyzed 4CzIPN-Mediated Decarboxylative Acylation Reaction of O-Methyl Ketoximes with α-Keto Acids under Visible Light"

_molecules, 2022, doi:10.3390/molecules27061980_

Round 1

Reviewer 1 Report

In this manuscript, the authors report a palladium-catalyzed oxime ether-directed ortho-selective benzoylation using benzoyl formic acid under visible light. The scope of various benzophenone derivatives was obtained in reasonable yields. This is a well-written and executed study worthy of publication. I only have a few suggestions/corrections for the authors to address:

  • In the abstract and conclusion part: it is better to call “non-symmetric” instead of “asymmetric” (page 1, line 13).
  • In the experiment section: The authors mentioned about Figure 2, however I didn’t see it in the manuscript. In fact, it is Figure 1 in the SI. So, they should put the figure in the main text or to give an appropriate reference to SI.
  • Some minor errors:

line 28, page 1, A dot should be after parenthesis of Figure 1a.

line 205, page 8 instead of was should be were

Reviewer 2 Report

The authors developed a new simple protocol for decarboxylative acylation of protected arylketoximes / preparation of the derived diarylketoximes using a combination of 10 mol % Pd(OAc)2 catalyst and 2 mol % 4CzIPN photocatalyst. After deprotection the reaction products furnish diaryketones that find numerous practical applications, so rendering this work practically valuable. A remarkable 'green' feature of the new protocol is that the only oxidant used here is O2 from air. Overall, the manuscript is well organized and I recommend acceptance after minor revisions mentioned below.

  • On p. 1, line 28, please correct the comment for Fig. 1-a. This reaction does not involve a-ketoacids.
  • On p. 2, lines 36-37 the authors state that the new protocol does not need an extra oxidant. That should be corrected since later in the text, in particular, based on mechanistic tests with DMPO, they argue that the actual terminal oxidant is O2. It might be better to state that the new protocol utilizes O2 as the only oxidant. To confirm that it would be important to include one more test in Table 1 for the reaction run under otherwise optimal conditions but in the absence of air (say, under argon atmosphere).
  • Accordingly, please add ‘air’ to each reaction scheme when characterizing reaction conditions for the new protocol: a) in Fig. 1-c, b) Scheme 2, c) Scheme 3, d) Scheme 4, e) Scheme 5.
  • On p. 9, lines 249 and 251 they provide a wrong formula, O2‑, for superoxide radical anion. A bit lower, lines 253-254, they say that “characteristic signal of O2-● was clearly observed”. Since DMPO, a radical trap, was actually used in the experiment, this statement is not correct and should be corrected.

Reviewer 3 Report

Huang, Zhao and co-workers developed the Pd-catalyzed decarboxylative acylation of O-methyl ketoximes with alpha-keto acids in the presence of photocatalyst under visible light irradiation. The reactions proceeded using Pd(OAc)2/4CzIPN in AcOH under air. Various substrates were converted to the corresponding products in moderate to high yields. Mechanistic insights were obtained by a reaction in the presence of TEMPO and KIE experiments. This was a nice touch. The manuscript was well-written. Overall, the present work is suitable for publication in Molecules as it is. 

Author Response

Thanks for your recognition for our work.